# GeneVLM: Automated Parsing Executable Digital Gene from a Single Image

## Abstract

Building structured world representations for robotic agents that can generalize and interact with the physical world is a core challenge in AI. The recently proposed *Digital Gene* offers a promising direction by representing objects as explicit, programmatic blueprints, addressing the generalization and interpretability bottlenecks of end-to-end learning paradigms. However, the practical application of this technology is hindered by a critical bottleneck: creating these genes for real-world objects. Prior methods rely on 3D data, which is difficult to acquire at scale, while parsing directly from ubiquitous 2D images remains an unsolved challenge.

In this work, we introduce **GeneVLM**, a vision-language framework that addresses this bottleneck by automatically parsing executable Digital Gene from a single 2D image. First, we propose a specialized and scalable model designed for the image-to-gene parsing task. Second, to enable its training, we design an efficient and scalable procedural pipeline to synthesize a diverse, multi-million-pair dataset of images and their corresponding Digital Genes. Third, to facilitate rigorous evaluation, we establish and release the first comprehensive, multi-dimensional benchmark for this task. Our experiments show that GeneVLM successfully recovers complex object structures and exhibits consistent performance scaling with increased model size, validating the effectiveness of our integrated approach.

## 1 Introduction

Recent advances in robotic learning, such as Vision-Language-Action Models (Black et al., 2024; Li et al., 2024; Kim et al., 2024a), reinforcement learning (Hafner et al., 2024; Hansen et al., 2022; Zhu et al., 2023), and diffusion policies (Ze et al., 2024; Chi et al., 2024), have shown great potential in training agents for complex robotics tasks. However, the dominant paradigm of end-to-end learning from raw sensory inputs, such as images, confronts fundamental bottlenecks. These models often require vast amounts of interaction data, struggle to generalize to novel objects or environments, and operate as "black boxes," making their behavior difficult to predict, interpret, or verify.

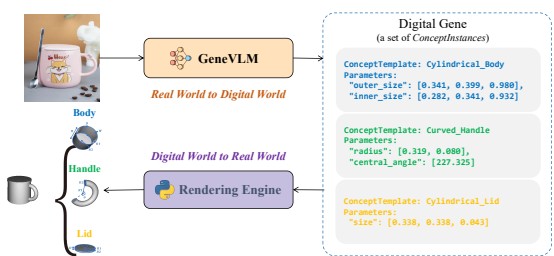

**Figure 1: Illustration of the Digital Gene.** The mug in the figure comprises three *ConceptInstances*, with their corresponding *ConceptTemplates* being **Cylindrical_Body**, **Curved_Handle**, and **Cylindrical_Lid**. GeneVLM parses the Digital Gene of the mug, which is then executed by the Rendering Engine to produce a 3D Mesh.

Recently, the concept of a **Digital Gene** (Sun & Lu, 2025) has been introduced. In contrast to traditional methods that rely on *implicit* representations, it uses an ***explicit, structured*** representation that introduces strong inductive biases aligned with the physical nature of manipulation tasks. A Digital Gene is an executable blueprint that encodes an object's hierarchical components, geometry, and functional attributes (e.g., joints, affordances). This hierarchical structure provides a powerful

abstraction for a robot to reason about how to grasp any mug, regardless of its specific dimensions, making it easier for the robot to generalize to unseen objects and rendering its behavior more interpretable and predictable.

However, the application of Digital Genes in real-world scenarios is limited by a key bottleneck: the annotation of objects in the physical world. While prior work has explored automatic generation from 3D point clouds (Sun et al., 2024c), this reliance on specialized sensors limits its use in uncontrolled, real-world environments where 2D images are the most ubiquitous and accessible sensing modality. Therefore, we argue that automatically parsing an object's Digital Gene directly from a single image is a crucial step towards the practical application of this technology and an important problem that needs to be addressed.

In this work, we address this critical challenge by proposing **GeneVLM**, a vision-language framework for the automated parsing of Digital Genes from single, 2D images. Our work makes the following contributions:

- **A Specialized Model for image-to-gene (GeneVLM):** We propose a scalable image-to-gene model. Our experiments show GeneVLM successfully recovers complex object structures and exhibits consistent performance scaling with increased compute.

- **An Efficient Data Generation Pipeline and a Large-Scale Public Dataset:** We design an efficient and scalable pipeline to synthesize a diverse, multi-million-pair dataset of images and corresponding Digital Genes. We use this dataset to successfully train GeneVLM and will release it publicly to facilitate future research in structured 3D understanding and robotic manipulation.

- **A Comprehensive Multi-Dimensional Benchmark:** We establish and release the first rigorous benchmark for the image-to-gene parsing task. It evaluates performance across three critical axes: (1) Gene-level matrics; (2) Geometric similarity; and (3) Perceptual similarity.

The remainder of this paper is organized as follows. We first introduce related works in Sec. 2 and the preliminary concepts of Digital Genes in Sec. 3. We then detail our proposed data generation pipeline in Sec. 4. Next, we introduce the GeneVLM model in Sec. 5. Subsequently, we present our comprehensive evaluation metrics and dataset in Sec. 6. Then, we present experiment setting, quantitative and qualitative results, and ablation study in Sec. 7. Finally, we conclude with a discussion of our findings and future work.

## 2 RELATED WORK

Understanding and representing the physical world for perception, reasoning, and control has been approached from two broad directions: *explicit*, interpretable descriptions grounded in geometry and mechanics, and *implicit*, learned representations whose structure is distributed in neural parameters. We review both families and position **Digital Gene** within this landscape in Append B.

## 3 PRELIMINARY

To understand the contribution of our work, it is essential to clearly define its target output. We build upon the notion of **Digital Genes**, a formal, programmatic framework for representing physical world concepts introduced by Sun & Lu (Sun & Lu, 2025). The original work defines Digital Genes at two levels of abstraction: high-level, executable Python classes called *Concept Templates* that define an analytic concept of an object (e.g., a general 'Cylindrical_Body' of a mug), and specific *Concept Instances* that represent a single analytic concept with fixed parameters.

As shown in Figure 1, when we refer to the Digital Gene of an object, we are specifically referring to the set of *Concept Instances* it comprises, represented as a parameterized JSON file. This JSON file is the direct output of our GeneVLM model and serves as a structured, machine-readable blueprint for a single object observed in an image. Crucially, this symbolic representation is not merely descriptive; it is *actionable*. As illustrated in the original Digital Gene pipeline (Sun & Lu, 2025), this JSON instance can be processed by a procedural generation engine to render a corresponding 3D mesh that contains manipulation knowledge. This explicit link from a structured text file to

a physical 3D model and its manipulation knowledge is what makes the Digital Gene a powerful representation for robotics.

# 4 DATASET

Training a model for the novel task of image-to-gene parsing requires a massive training dataset of paired images and their corresponding Digital Genes. Manually creating the required large-scale dataset of paired images and Digital Genes is unscalable. We therefore developed a procedural pipeline to automatically generate diverse, high-fidelity data by systematically varying object structure, pose, and visual realism, as illustrated in Figure 2. The entire process yields two final datasets: a large-scale *WhiteImage-Gene* Dataset for learning Digital Gene priors, and a smaller, higher-fidelity *ColorImage-Gene* Dataset designed to bridge the simulation-to-reality gap.

**Digital Gene Database.** The pipeline begins with a seed collection of Digital Genes, covering distinct object categories such as chairs and mugs. We programmatically synthesizes new gene variations (Sun et al., 2024a) by: (1) substituting high-level *ConceptInstance* components within a gene's template (e.g., swapping one leg style for another); (2) adjusting discrete parameters (e.g., modifying the number of buttons on a controller); and (3) sampling new values for continuous parameters (e.g., altering object dimensions). This automated expansion rapidly populates our Digital Gene collection with structures of rich combinatorial diversity. We then introduce two key diversification stages to mitigate the inherent bias of synthetic data and improve the diversity of our dataset. First, to enhance object diversity, we stochastically remove non-essential *ConceptInstance* from the genes. The rationale is that real-world objects often deviate from their idealized forms; for example, a mug may be missing its lid. Second, to ensure diversity in the 6D pose, we apply pose augmentation. Specifically, for each gene, we generate five distinct 6D poses by sampling a 'position(x, y, z)' within [-1.0, 1.0] on each axis and a 'rotation (x, y, z) ' within [-180, 180].

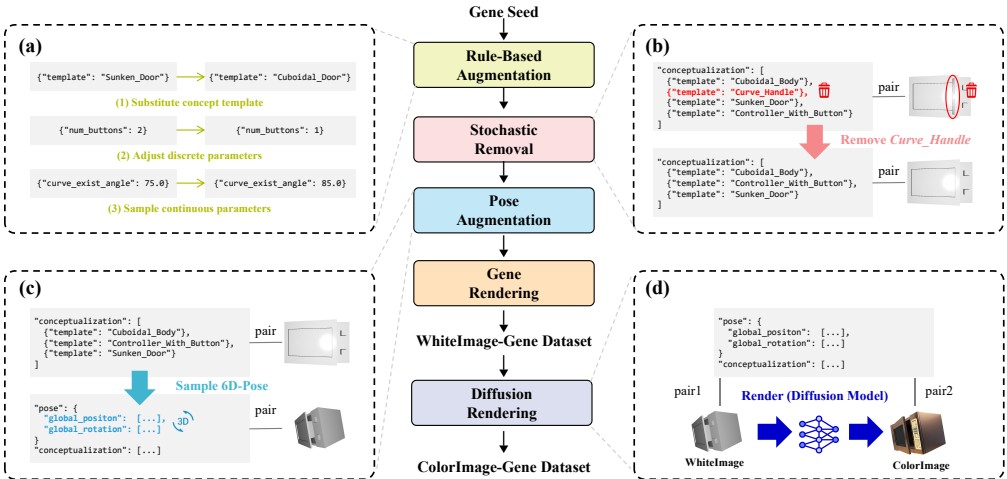

**Figure 2:** An overview of our data generation pipeline. (a) showcases rule-based gene synthesis. (b) demonstrates the stochastic removal of a non-essential concept, *Curve_Handle*. (c) illustrates pose augmentation applied to the gene. (d) shows the use of a diffusion model to add color and texture to a *WhiteImage*, resulting in a *ColorImage*. For clarity in the diagram, we have also visualized the genes at stages before Gene Rendering to more intuitively demonstrate the effect of our operations.

**Image-Gene Dataset.** After obtaining a diverse and augmented dataset of Digital Genes, we proceed to the rendering phase to generate *Image-Gene* pairs. We first generate the large-scale **WhiteImage-Gene Dataset**. This is done by passing each gene through the rendering engine to produce a 3D mesh. The mesh is then rendered from a fixed camera perspective into a 2D image that contains only the 3D mesh, without any texture, lighting, or material. Since the object in this 2D image appears grayish white, we refer to it as a *WhiteImage*. This step is computationally efficient, allowing us to generate a vast corpus covering a wide range of structures and poses. It

serves as the primary data source for teaching the model the fundamental mapping from visual geometry to gene. To bridge the gap between *WhiteImages* and real photos, we leverage a pretrained diffusion model (Team, 2025b) to transform them into more realistic images, thereby creating the **ColorImage-Gene Dataset**. While preserving the underlying mesh structure, this process adds factors such as color, lighting, and texture. This stage is more computationally intensive, resulting in a smaller but higher-quality dataset that is essential for fine-tuning the model to generalize to real-world visual inputs.

# 5 GENEVLM MODEL

## 5.1 MODEL ARCHITECTURE

Our model, which we term GeneVLM, is built upon the Qwen2.5-VL (Team, 2025a) architecture and fine-tuned for the specialized task of Digital Gene parsing. A significant challenge during fine-tuning arises from the specific structure of *ConceptInstance*, which consist of key-value pairs where values are often high-precision floating-point numbers. Standard tokenizers lead to excessively long token sequences, a verbosity that not only severely degrades training and inference efficiency, but also obscures the overall gene structure, making it difficult for the model to predict the correct gene. To overcome this critical bottleneck, we introduce a specialized tokenization scheme for numerical values, inspired by Open-VLA (Kim et al., 2024b).

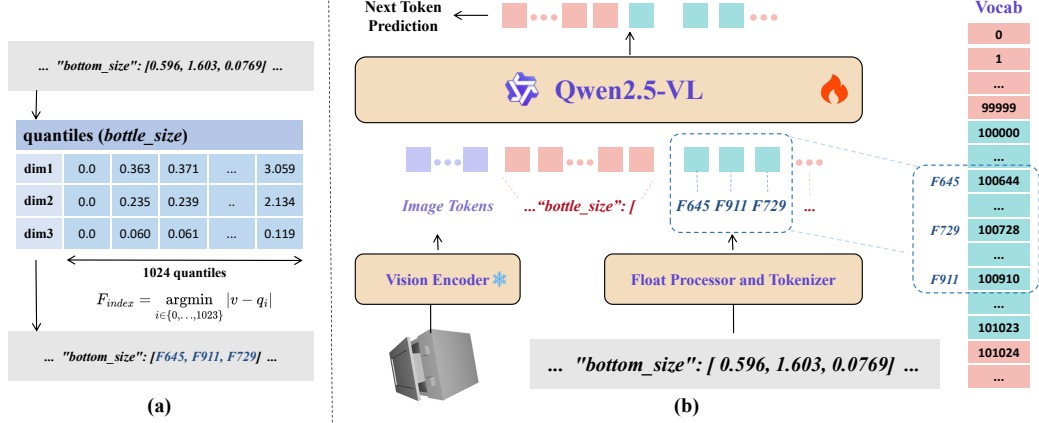

**Figure 3:** Overview of our model architecture. (a) illustrates the preprocessing of floating-point numbers to obtain $F_{index}$. (b) shows the overall model architecture and illustrates how $F_{index}$ is mapped into the vocabulary space.

As illustrated in Figure 3, our method converts each floating-point number in a gene into a single special token, thereby preserving its semantic integrity and drastically shortening the sequence length. Specifically, for each key in the *ConceptInstance*, we first analyze its value distribution within the dataset and pre-compute 1024 quantiles. During data preprocessing, any given value is mapped to the index of its nearest pre-computed quantile. Formally, given a raw value $v$ and an ordered set of 1024 quantiles $\{q_0, q_1, \ldots, q_{1023}\}$, its index $F_{index}$ is calculated as:

$$F_{index} = \operatorname*{argmin}_{i \in \{0,\ldots,1023\}} |v - q_i| \tag{1}$$

Finally, we allocate a rarely used space in the vocabulary (starting at BoF) to map these indices:

$$F_{index} \to \text{BoF} + F_{index} \tag{2}$$

This quantization strategy offers several profound advantages. Firstly, it drastically reduces the sequence length for each data sample, leading to a significant reduction in computational cost and a corresponding increase in training throughput. Secondly, this atomic representation allows the model to treat numerical parameters as single semantic units, enabling it to focus on the high-level syntax and structure of the Digital Gene. Furthermore, this discrete representation greatly simplifies the implementation of the constrained decoding logic used during inference. Crucially, this process also implicitly embeds a strong prior of the dataset's numerical distribution into the system, guiding the model towards generating plausible parameter values.

We employ a two-stage training strategy to effectively leverage our generated datasets. The first stage focuses on learning the foundational mapping from geometry to gene, where the model is trained on the *WhiteImage-Gene* dataset. The second stage is dedicated to bridging the sim-to-real gap, where the model is fine-tuned on the *ColorImage-Gene* dataset. We mix in a general-purpose SFT dataset during both stages to ensure that the model retains its powerful capabilities for general visual understanding. Throughout both stages, the vision encoder is kept frozen to preserve its image encoding capabilities.

## 5.2 CONSTRAINED DECODING

Standard auto-regressive decoding often fails to generate syntactically valid Digital Genes, as their strict structure is easily violated. Such errors render the output gene unexecutable and unusable. To overcome this challenge, we introduce a constrained decoding method to guarantee the executability of the genes predicted by our model.

At each sampling step, based on the sequence generated so far and the strict constraints of the digital genes themselves, we identify a set of permissible tokens for the current step and mask out all others. This forces the model to sample only from a grammatically valid set, effectively guiding the generation process. Our constrained decoding method is implemented as a Finite State Machine (FSM) where each state corresponds to a specific point in the Digital Gene's abstract syntax tree. As a result, the reliability and utility of our model are significantly improved. A detailed implementation and its pseudocode are provided in Appendix A.2.

## 6 EVALUATION

We design a multi-dimensional benchmark and set of metrics to comprehensively evaluate the GeneVLM performance.

### 6.1 GENE-LEVEL EVALUATION

To quantitatively evaluate the generated Digital Genes, we established two primary metrics: Concept Accuracy and Float Error. These are assessed on a test set of 15k *WhiteImage-Gene* pairs.

**Concept Accuracy.** This metric evaluates the rate of structural correctness for the genes generated by the model over the entire test set. A generated gene, $G_{pred}$, is defined as structurally correct with respect to its ground truth, $G_{gt}$, if and only if it satisfies all the conditions outlined in the following expression:

$$\text{IsCorrect}(G_{pred}, G_{gt}) \iff \begin{cases} \text{IsExecutable}(G_{pred}) & \wedge \\ T(G_{pred}) = T(G_{gt}) \end{cases} \tag{3}$$

where $T(G)$ represents the set of the gene's *ConceptTemplates*.

**Float Error** This metric is calculated only for genes that are deemed structurally correct and evaluates the precision of its numerical parameters. To account for the varying scales and meanings of different parameters, we calculate the Mean Absolute Error on their corresponding quantile indices (as detailed in Section 5.1). This approach normalizes the errors onto a unified scale of 0-1024, allowing for a consistent and meaningful comparison.

### 6.2 GEOMETRIC AND PERCEPTUAL EVALUATION

For images that do not have a reference Digital Gene, we design a comprehensive, two-part benchmark that jointly measures (i) **geometric similarity** under standardized 3D metrics (e.g., Chamfer distance and F-scores), and (ii) **perceptual similarity** via a *VLM-as-a-judge* protocol. We adopt this dual-aspect evaluation because each metric possesses complementary strengths. While geometric metrics provide precise quantitative scores, they can be sensitive to misalignment and fail to capture semantic plausibility in real scenes. Conversely, perceptual judgments are more robust in the wild but lack absolute geometric guarantees. By triangulating these two axes, our benchmark provides a principled evaluation, ensuring that the generated Digital Genes are not only geometrically faithful but also produce visually coherent results that align with human perception.

### 6.2.1 GEOMETRIC EVALUATION

**Dataset and setup** As mentioned in Sec. 3, the Digital genes generated by GENEVLM could be converted to mesh. Thus, GENEVLM could be considered as single-image 3D reconstruction model as well. Thus, we follow established single-image evaluation protocols from recent works (Liu et al., 2025; Huang et al., 2023; Liu et al., 2023b) to establish this benchmark. We construct our test set from the OmniObject3D dataset (Wu et al., 2023), selecting categories that overlap with our taxonomy. For each object, we uniformly sample 8 distinct views from the corresponding video, creating a set of 6,000 image-mesh pairs. The detailed breakdown of the number of samples in each class is provided in Appendix A.7.

**Metrics** We report two standard metrics for 3D shape comparison: **Chamfer Distance (CD)** and **F-Score**. CD measures the average squared distance between two point clouds, providing a measure of overall shape alignment. Given a predicted point cloud $P$ and a ground-truth point cloud $Q$, the symmetric CD is:

$$\mathrm{CD}(P,Q) = \frac{1}{|P|} \sum_{p \in P} \min_{q \in Q} \|p - q\|_2^2 + \frac{1}{|Q|} \sum_{q \in Q} \min_{p \in P} \|q - p\|_2^2. \qquad (4)$$

The **F-Score** provides a complementary view by measuring the harmonic mean of precision and recall at a given distance threshold $d$, indicating the percentage of the surface reconstructed within that tolerance. We report F@$\{0.01, 0.02, 0.05\}$, following common practice. To ensure fair evaluation of metrics under unknown global transforms, we adopt the commonly used alignment between predicted mesh and ground truth mesh (Huang et al., 2023). The details are provided in Appendix A.8

### 6.2.2 PERCEPTUAL EVALUATION

We also build a benchmark of 2,640 real-world images to test our GeneVLM in real-world scenarios (see Appendix for examples). This dataset has no 3D ground truth available; we assess semantic and geometric agreement by using a VLM to compare the input photo with renderings of the predicted mesh. We use Gemini2.5-flash (Comanici et al., 2025) for this purpose.

The complete evaluation pipeline is summarized as follows. Given the predicted mesh, we render multi-view RGB images using Blender. We sample views by rendering the object from eight fixed, equidistant azimuthal viewpoints, with the orientation set to look at the object center. From these eight sampled views, we randomly select four to form a (photo, renderings) pair for a single evaluation procedure. Each (photo, renderings) pair is scored by a fixed VLM prompt that asks for a scalar geometric-similarity judgment (the prompt is provided in Appendix A.3). We repeat this process for three independent runs and report the mean VLM score across the three runs. We provide examples of real images in this benchmark, corresponding predicted 3D mesh, and VLM similarity score in Appendix A.11.

**The validity of VLM-based similarity score** As this VLM-based similarity is newly introduced, its validity require evaluation. We validate this pipeline using a human-labeled partial-order set comprising 3,000 instances. The result shows this evaluation protocal achieving 92% agreement on a partial-order ranking task. This high consistency confirms the score's reliability and strong alignment with human perception. The detailed validation procedure is described in Appendix A.9.

## 7 EXPERIMENTS

### 7.1 SETUP

Our model is built upon the Qwen-2.5-VL. The model was trained in two stages. The first stage trained on 6M *WhiteImage-Gene* pairs. The second stage used a dataset of 800k *ColorImage-Gene* pairs. In both stages, we mixed in approximately 300k samples from three general SFT datasets (InternVL-SFT (Chen et al., 2023), ShareGPT-4o (Cui et al., 2024), and LLaVA (Liu et al., 2023a)) to ensure the model's ability to recognize and analyze real images.

**Baseline** We tested Qwen-2.5VL-32B as our baseline. Since Qwen has not been trained on our task, we wrote a detailed prompt explaining the gene's meaning, format, and examples to guide its generation. Additionally, a portion of the genes generated by Qwen were not executable, so we calculated the evaluation results on the correct samples to serve as the baseline. We report the prompt used for baseline in the Appendix A.4.

## 7.2 MAIN RESULTS

This section will present our main experimental results. We first showcase qualitative results to provide an intuitive demonstration of the model's capabilities, followed by a quantitative evaluation.

### 7.2.1 QUALITATIVE RESULTS

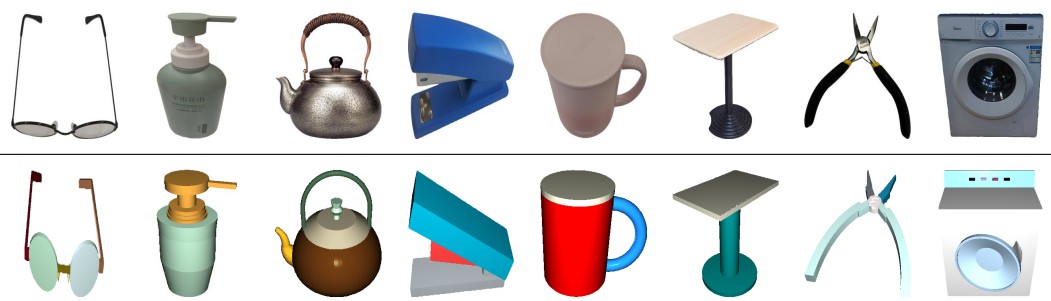

**Figure 4:** General results. The first row shows the input images containing the object to be parsed. The second row displays the visualizations rendered from the model's parsed Digital Genes. Different colors on the components represent different *ConceptImstances* defined in the gene.

**General Parsing Ability.** Figure 4 displays visualization results for 8 objects. From these visualizations, we can observe two key points. First, the model adeptly parses the fundamental structure and constituent components of the objects. This success is attributable to the robust visual recognition ability from its pre-training and the parsing capability endowed by our extensive Digital Gene dataset. Second, the model demonstrates the ability to parse a variety of everyday objects across multiple categories, successfully resolving the structure and functional components for all eight classes shown in the figure.

**Generalization Parsing Ability.** We conducted an experiment using eyeglasses to test the model's generalization ability. We captured multiple photos under varying camera viewpoints and lighting conditions while adjusting the open/-closed state of the eyeglass-leg. The results, shown in Figure 5, indicate that the model correctly identifies the object's constituent structure under these different conditions, demonstrating its robustness and that it is not limited to a specific perspective. Furthermore, the model is able to correctly extract and parse different object poses arising from its own joint articulation. This is proven by the fact that the open/closed state of the eyeglass-leg in the visualized results is consistent with the original images.

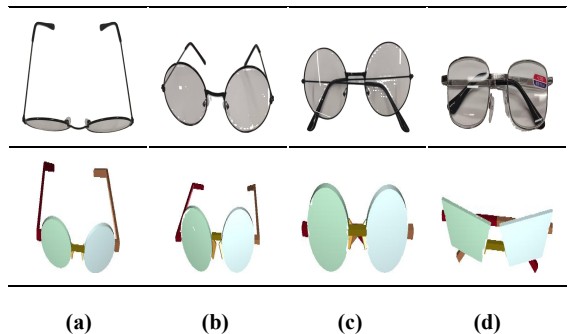

**Figure 5:** Generalization results. From the original image (a), we derive variations by adjusting the viewpoint and lighting (b), the object's articulated pose (c), and the specific instance within the same category (d).

### 7.2.2 QUANTITY RESULTS

Our main quantitative results are summarized in Table 1. A key finding is that our specially fine-tuned GeneVLM models substantially outperform the powerful Qwen-32B baseline. And our approach exhibits excellent scalability. Scaling the model from 7B to 32B parameters yields consistent performance improvements across all metrics. This positive scaling trend indicates that GeneVLM is a robust and promising framework, offering a scalable solution to the challenging 'image-to-gene' task.

To provide a reference for future research, the scaling properties of the proposed model were further investigated. Specifically, the relationship between training computation, measured in GFLOPS, and various evaluation metrics was tracked using the Gene-VLM-7B training configuration. As

| Model | Concept Acc. | Float Err. ↓ | VLM-Score | CD ↓ | F@0.01 | F@0.02 | F@0.05 |
|---|---|---|---|---|---|---|---|
| Qwen-32B | 0.112 | 232.205 | 0.402 | 0.131 | 0.051 | 0.271 | 0.3459 |
| Gene-7B | 0.625 | 122.46 | 0.721 | 0.111 | **0.1523** | 0.2344 | 0.5013 |
| Gene-32B | **0.660** | **115.367** | **0.815** | **0.055** | 0.1385 | **0.3147** | **0.6086** |

**Table 1:** Main results. Arrows indicate preferred direction for metrics (↓ lower is better).

depicted in Figure 6a, both Concept Accuracy and Float Error demonstrate improvement with increased computational resources, which provides evidence of predictable code-generation scaling. Furthermore, Figure 6b illustrates a generally monotonic increase in the VLM-Score across the observed computational budgets, with no apparent plateau within the tested range. Collectively, these results indicate that greater computational investment yields enhanced performance in structure extraction, and stronger geometric similarity.

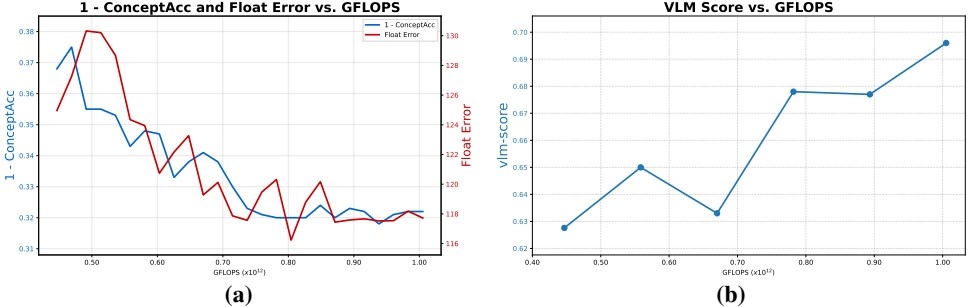

| (a) | (b) |
|---|---|

**Figure 6:** Scaling Properties. (a). Concept Accuracy and Float Error. Both 1 - ConceptAcc and Float Error decrease as the GFLOPS increases, demonstrating a clear scaling trend. (b) VLM-Score. The VLM-Score increases with GFLOPS, showing consistent gains. We report the detailed data in the Appendix A.6.1

## 7.3 ABLATION STUDIES

### 7.3.1 EFFECT OF SPECIALIZED NUMERICAL TOKENIZATION

We conducted an ablation study to evaluate our specialized tokenization scheme for floating-point numbers. We trained two models on a dataset of 400k white-background images across eight object categories: **Model-B** (base), which uses a standard tokenizer, and **Model-S** (special), which employs our proposed Float Tokenization method.

First, we analyzed the impact of our method on the sequence length of the training data. Figure 7 shows the sequence lengths for the eight categories before and after applying our tokenization scheme. It is evident that our method reduces the sequence length by nearly half in every category, which implies a significant reduction in model training costs and a substantial increase in inference speed.

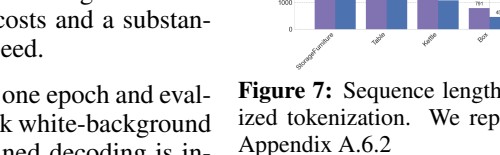

We trained both models for one epoch and evaluated them on a test set of 4k white-background images. Since our constrained decoding is incompatible with Model-B, we used standard auto-regressive decoding for both models.

**Figure 7:** Sequence lengths with and without specialized tokenization. We report the detailed data in the Appendix A.6.2

| Model | Training PFLOPS | Inference Hours | Concept Accuracy ↑ | Float Error ↓ |
|---|---|---|---|---|
| Model-B | 52719 | 5.5 | 0.568 | **132.912** |
| Model-S | 26355 | 1.5 | **0.668** | 134.176 |

**Table 2:** Ablation study on specialized tokenization scheme.

The data in Table 2 shows that, compared to Model-S, Model-B requires significantly more training compute for the same amount of raw data and performs markedly worse in both inference speed and result quality. This fully demonstrates the effectiveness of our method. At the same time, we note that the accuracy of parameter prediction is comparable between the two models, with Model-S showing no clear advantage. Therefore, this operation does not directly improve the model's precision in predicting parameters, an aspect we will explore in future work.

### 7.3.2 Effect of General-Purpose SFT Data

We conducted an ablation study on the general-purpose SFT dataset using a smaller training set. This set contained 750k white-background images and 750k color images from 15 object categories. We trained three models: (1) **Model-W**, using only *WhiteImage-Gene* data pairs; (2) **Model-WC**, using both *WhiteImage-Gene* and *ColorImage-Gene* data pairs; and (3) **Model-WG**, using *WhiteImage-Gene* data pairs mixed with the general SFT dataset. We evaluated them on a dataset of 1,000 real-world images, with results shown in Table 3.

| Model | CD $\downarrow$ | F@0.01 $\uparrow$ | F@0.02 $\uparrow$ | F@0.05 $\uparrow$ |
|---|---|---|---|---|
| Model-W | 0.35885 | 0.079454 | 0.190559 | 0.41414 |
| Model-WC | 0.11184 | **0.152324** | 0.234478 | 0.50139 |
| Model-WG | **0.06109** | 0.100066 | **0.24722** | **0.51837** |

**Table 3:** Ablation study on the effect of a general-purpose SFT dataset.

The results clearly demonstrate that incorporating a general-purpose SFT dataset is effective for improving performance on real-world images. We conclude that this is due to the significant domain gap between our synthetic white-background images and real images. Training on large-scale synthetic data alone can degrade the model's ability to understand real images, and mixing in the SFT dataset effectively mitigates this problem. We also note that Model-WC underperforms Model-WG in this experiment. We speculate this may be due to quality issues within the unfiltered 750k color images, which could have negatively impacted the model's performance.

## 8 Limitations and Future Work

Despite the promising results, our work has several limitations.

First, during the rendering of *ColorImage* from *WhiteImage* using the pretrained diffusion model, unavoidable alterations to the mesh structure can occur. This may introduce noise into the *ColorImage-Gene* dataset, potentially affecting the model's learning.

Second, the model's current prediction accuracy may not yet meet the high-precision requirements for real-world robotic manipulation.

Finally, our evaluation metrics could be refined. The gene-level metrics may not capture fine-grained errors in the generated genes. Concurrently, the VLM-as-a-Judge evaluation can be susceptible to the inherent biases of the VLM itself.

## 9 Conclusion

This paper addresses the critical manual annotation bottleneck in the creation of Digital Genes by proposing an automated solution named GeneVLM. Through an innovative reverse-synthesis data pipeline, a robust vision-to-program translation model, and a constrained decoding technique that guarantees syntactic validity, the GeneVLM framework successfully achieves end-to-end conversion from visual diagrams to executable Digital Gene code. We designed and implemented a comprehensive evaluation suite, including a novel VLM-as-a-Judge metric, and our experimental results fully demonstrate the effectiveness and superiority of our method. This work not only paves the way for the large-scale application of Digital Genes but also provides a crucial technical foundation and a new research paradigm for advancing AI systems toward a deeper and more reliable understanding of and reasoning about the physical world.

## 10 REPRODUCIBILITY STATEMENT

Reproducibility Statement. We aim to make our results fully reproducible by referencing where all necessary details appear in the paper and supplement. Data generation: Sec. 4 and Fig. 2 document the complete pipeline (rule-based gene synthesis, stochastic component removal, pose augmentation, and White/Color image rendering), and category counts for the geometric benchmark are listed in App. A.7. Model & training: Sec. 5.1 specifies the GeneVLM architecture and float-tokenization scheme (with equations), while Sec. 5.2 and App. A.2 provide the constrained-decoding FSM and pseudocode; the exact training output format/prompt is in App. A.5, and the two-stage training setup and dataset sizes are summarized in Sec. 7.1. Evaluation: Sec. 6.1 defines Concept Accuracy and Float Error; Sec. 6.2.1 details the geometric metrics (Chamfer Distance and F-score, with Eq. (4)) and the mesh-alignment protocol is in App. A.8; Sec. 6.2.2 outlines the VLM-as-judge procedure with the precise prompt (App. A.3) and its human-agreement validation (App. A.9). We further report per-budget scaling data and sequence-length statistics in App. A.6, and include the baseline prompting used for comparisons in App. A.4. In the anonymous supplementary materials, we will provide a downloadable code archive containing model implementation, constrained decoding ,and evaluation benchmark sufficient to reproduce all tables and figures.

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

# A APPENDIX

## A.1 USE OF LARGE LANGUAGE MODELS (LLMs)

In accordance with the ICLR policy on LLM usage, we disclose that we used an LLM (ChatGPT, OpenAI) only after completing the full manuscript draft, and solely for surface-level proofreading: correcting grammar, punctuation, and minor phrasing for clarity and consistency. The LLM did not contribute to research ideation, problem formulation, method or experiment design, data collection or labeling, analysis. Every suggested edit was manually reviewed and selectively adopted by the authors.

We understand and accept full responsibility for all content written under our names, including any text that may have been revised with LLM assistance. We took care to avoid plagiarism and factual errors and did not provide the LLM with proprietary or personally identifiable data beyond de-identified manuscript excerpts necessary for proofreading. The LLM is not an author or contributor under ICLR authorship criteria.

## A.2 DETAILED IMPLEMENTATION OF CONSTRAINED DECODING

To enforce the grammatical integrity of the output at inference time, we employ a deterministic Finite State Machine (FSM) to guide the decoding process. This FSM acts as a grammar-aware guardrail, ensuring that every generated token conforms to the strict syntax of a valid Digital Gene.

Our FSM is composed of a set of discrete states (e.g., *GEN_CATEGORY_VALUE*, *ADD_POSITION_KEY*, *GEN_PARAM_VALUE*), where each state corresponds to a specific node or element in the hierarchical JSON structure of the Digital Gene. The generation process starts in an initial state and transitions between states based on a predefined transition table, effectively traversing the abstract syntax tree of the gene. This mechanism is implemented via a custom 'LogitsProcessor' within the generation pipeline, as detailed in Algorithm 1.

---

**Algorithm 1** FSM-Guided Constrained Decoding

---

1: **Input:** Model logits $S \in \mathbb{R}^V$, generated token sequence $I_{gen}$, FSM state $F_{state}$
2: **Output:** Masked logits $S'$
3: **function** PROCESSLOGITS($S, I_{gen}, F_{state}$)
4:     $F_{state} \leftarrow$ UPDATEFSMSTATE($I_{gen}$)
5:     $V_{allowed} \leftarrow$ GETALLOWEDTOKENS($F_{state}$)
6:     $M \leftarrow \text{ones}(V) \times (-\infty)$             ▷ Create a mask with $-\infty$ for all tokens
7:     **for** $v \in V_{allowed}$ **do**
8:         $M[v] \leftarrow 0$         ▷ Unmask allowed tokens by setting their mask value to 0
9:     **end for**
10:     $S' \leftarrow S + M$            ▷ Apply mask to original logits
11:     **return** $S'$
12: **end function**

---

At each generation step, the processor intercepts the model's output logits. Based on the FSM's current state, it identifies a small subset of permissible next tokens. The logits for all other tokens in the vocabulary are masked (set to $-\infty$), forcing the model to sample only from the valid set. These constraints operate in two modes: for deterministic syntactic elements (e.g., keys like '"pose":"' or structural characters like '['), the FSM forces the exact token sequence; for generative content (e.g., a template name or a parameter value), the FSM restricts the output to the class of valid tokens

(e.g., any of the known template names or the special quantized numerical tokens introduced in our tokenization scheme).

This FSM-guided approach guarantees the syntactic correctness of the output by construction, completely eliminating the possibility of structural errors and significantly improving the executability rate of the generated code. It transforms the generation task from a purely probabilistic sequence prediction into a structured traversal problem, leveraging the VLM's powerful visual understanding to make informed choices within a grammatically sound framework.

## A.3 PROMPT USED FOR VLM-SCORE QUERY

Here is the prompt that we used in the VLM-score query described in Sec 6.2.2.

You are a 3D geometry comparison expert. I will give you a colored image and four rendered images. Four rendered images are from different viewpoints of a same object.
Compare the geometric shape of the object in the colored image with the objects in the four rendered images provided. *Do not consider texture or color differences.* Focus *exclusively* on the 3D shape, proportions, and the presence, absence, and relative positioning of components.
First, describe your reasoning step-by-step. Analyze the similarities and differences you observe between the colored image and the rendered images. Consider:
* Overall shape and silhouette.
* Presence and relative position of major components.
* Proportions and sizes of components.
* Any noticeable distortions, exaggerations, or omissions.
* Specific features and details.

After your detailed reasoning, provide a single numerical score between 0.0 and 1.0, representing the geometric similarity. Use the following scale as a guide:
* **1.0:** Perfect geometric match. All aspects of the 3D shape are identical.

* **0.9 - 0.99:** Near-perfect match. Minor differences, possibly in fine details or slight proportional variations that are barely perceptible.
* **0.8 - 0.89:** Very good match. The overall structure is the same, but there might be small, noticeable differences in the size, shape, or angle of some sub-components. The core geometry is preserved.
* **0.7 - 0.79:** Good match. The general shape is recognizable, but there are clear differences in several sub-components. Some features might be slightly exaggerated, compressed, or otherwise distorted.
* **0.6 - 0.69:** Moderate match. The object is still identifiable, but significant differences are present. The arrangement of some sub-components might be altered, or their shapes might be substantially different.
* **0.5 - 0.59:** Fair match. The basic silhouette might be similar, but major structural differences are evident. This might involve missing or added components, or significant changes in component placement.
* **0.4 - 0.49:** Poor match. Only a vague resemblance remains. Key structural elements are different or missing. The object's overall form is substantially altered.
* **0.3 - 0.39:** Very poor match. Minimal resemblance. Major components are missing, added, or drastically changed.
* **0.2 - 0.29:** Extremely poor match. Almost no geometric similarity.
* **0.0 - 0.19:** No discernible geometric similarity. * if there are any dis-connected sub-components or un-reasonable sub-components(assume all objects are daily objects) in the rendered images, the score should be lower than 0.3.

Your final answer *MUST* end with a line in the following format:

`FINAL SCORE: X.X`

Where `X.X` is the numerical score (e.g., `FINAL SCORE: 0.8`). The reasoning should come *before* this line.

## A.4 Prompt used for Baseline

Here is the prompt that we used in the Qwen generation procedur described in Sec 7.1.

You are an image-to-JSON scene encoder.
The user supplies **one image**.
You MUST output **only** a JSON object that lists every recognizable object in the image, following the schema below.

---

### Scene JSON schema

```
{
"category":  "<One of the top-level keys in param_dims.py>",
"pose":  {
"global_position":  [value_x, value_y, value_z], // object's
offset
"global_rotation":  [value_x, value_y, value_z] // rotation
order is x -> y -> z, range:  [-180, 180]
},
"conceptualization":  [
{
"template":  "<One legal concept name under that category
(param_dims.keys())>",
"parameters":  {
"<param_1>":  [ value_1, value_2, ...  ], // length must
match the vector length
"<param_2>":  [ ...  ],
...
}
},
{
next template ...
}
]
}
```

here is param_dims.py:
```python
param_dims = {param_str}
```

- **template and parameter names/vector lengths are authoritative in param_dims.py**.
  - A value of '[2]' means exactly two numbers, '[3]' means three numbers, '[2,3]' means two **or** three numbers are acceptable.
- Omit any parameter that is **not** listed for the chosen concept.
- Use **lower-case decimal numbers** (floats). Units are metres for lengths/positions and degrees for rotations unless the parameter's meaning implies otherwise.
- Put objects in the '"objects"' array **in any order**; each physical part (e.g. a mug body and its handle) is a separate object entry.
- Output **only** valid JSON – no comments, no trailing commas, no additional keys, no explanatory prose.

### Tips & constraints

1. **Vector length correctness is critical.** If a parameter's required length $\neq$ the length you output, the scene will be rejected.

2. **Spell everything exactly** as in 'param_dims.py' (case-sensitive).

3. If an object is partially occluded, estimate its parameters from visible evidence.

### Example:

```

{example_str}

```

## A.5 PROMPT USED FOR TRAINING

Here is the prompt that we used in training described in Sec 5.1.

> You are given a task that involves both language reasoning and image understanding. Based on the provided textual and visual inputs, estimate the underlying structure and parameters of the described object. Your goal is to generate a structured representation of the object as JSON code.
>
> Use both linguistic reasoning and visual cues to infer the object's geometry, configuration, and relevant parameters.
>
> All numerical values in the code should be linearly mapped and discretized into integers within the range 2048 to 3072.
>
> The final output must be a JSON code block enclosed within  and  tags. Only include the code inside these tags — no explanations, descriptions, or formatting outside of them.
>
> Ensure your output is accurate, complete, and strictly adheres to this format.

## A.6 DETAILED EXPERIMENTAL RESULTS

### A.6.1 DETAILED DATA FOR SCALING ANALYSIS

The following Table 4 provides the detailed data for the scaling law analysis presented in the main text. It records the changes in Gene-level evaluation metrics as the training computation (GFLOPS) increases.

The following Table 5 provides the detailed data for the VLM-Score, tracking its change as the training computation (GFLOPS) increases.

### A.6.2 DETAILED DATA FOR SEQUENCE LENGTH REDUCTION

The following Table 6 provides a detailed breakdown of the sequence length before and after applying our specialized tokenization scheme for floating-point numbers. The data supports the analysis of training efficiency gains discussed in the main text.

| GFLOPS ($\times 10^{12}$) | Concept Acc | Float Error |
|:---:|:---:|:---:|
| 0.45 | 0.632 | 124.962 |
| 0.47 | 0.625 | 127.259 |
| 0.49 | 0.645 | 130.322 |
| 0.51 | 0.645 | 130.203 |
| 0.54 | 0.647 | 128.666 |
| 0.56 | 0.657 | 124.347 |
| 0.58 | 0.652 | 123.945 |
| 0.60 | 0.653 | 120.737 |
| 0.63 | 0.667 | 122.165 |
| 0.65 | 0.662 | 123.267 |
| 0.67 | 0.659 | 119.279 |
| 0.69 | 0.662 | 120.122 |
| 0.71 | 0.670 | 117.868 |
| 0.74 | 0.677 | 117.570 |
| 0.76 | 0.679 | 119.470 |
| 0.78 | 0.680 | 120.317 |
| 0.80 | 0.680 | 116.236 |
| 0.83 | 0.680 | 118.790 |
| 0.85 | 0.676 | 120.165 |
| 0.87 | 0.680 | 117.446 |
| 0.89 | 0.677 | 117.590 |
| 0.92 | 0.678 | 117.667 |
| 0.94 | 0.682 | 117.524 |
| 0.96 | 0.679 | 117.543 |
| 0.98 | 0.678 | 118.185 |
| 1.01 | 0.678 | 117.722 |

**Table 4:** Gene-level metrics as a function of training computation (GFLOPS).

| GFLOPS ($\times 10^{12}$) | VLM-Score |
|:---:|:---:|
| 0.45 | 0.6276 |
| 0.56 | 0.6500 |
| 0.67 | 0.6330 |
| 0.78 | 0.6780 |
| 0.89 | 0.6770 |
| 1.01 | 0.6960 |

**Table 5:** VLM-Score as a function of training computation (GFLOPS).

| | Min Tokens | | Max Tokens | | Average Tokens | |
|---|:---:|:---:|:---:|:---:|:---:|:---:|
| **Category** | **wo** | **w** | **wo** | **w** | **wo** | **w** |
| Bottle | 621 | 345 | 644 | 356 | 633.923 | 354.202 |
| Box | 568 | 316 | 1035 | 601 | 791.240 | 457.674 |
| Bucket | 324 | 210 | 614 | 376 | 419.169 | 263.602 |
| Dispenser | 410 | 314 | 726 | 437 | 566.430 | 371.812 |
| Kettle | 1353 | 855 | 2422 | 1411 | 1842.810 | 1088.989 |
| KitchenPot | 362 | 229 | 1074 | 663 | 714.620 | 456.029 |
| StorageFurniture | 5091 | 3022 | 10021 | 5303 | 7169.164 | 4024.182 |
| Table | 520 | 375 | 6962 | 4262 | 2432.800 | 1645.236 |

**Table 6:** Sequence length comparison with and without ("w/o" vs "w") specialized tokenization.

## A.7 CLASS DISTRIBUTION OF GEOMETRIC SIMILARITY BENCHMARK

| Category | Bottle | Box | Bucket | Chair | Eyeglasses | Kettle | Knife |
|---|---|---|---|---|---|---|---|
| Number | 500 | 820 | 560 | 580 | 100 | 240 | 560 |
| Category | Laptop | Microwave | Mug | Shampoo | Storage Furniture | Table | Trashcan |
| Number | 80 | 160 | 740 | 660 | 140 | 460 | 400 |

## A.8 ALIGNMENT PROCEDURE OF TWO MESH

To ensure fair evaluation of metrics under unknown global similarity transforms, we adopt the alignment protocol proposed in (Liu et al., 2025). Specifically, we first sample 10,000 points from the surfaces of the predicted and ground-truth meshes. Both point clouds are normalized to fit within a unit sphere. We then perform a coarse grid search over rigid rotations, followed by a fine-grained Iterative Closest Point (ICP) alignment.

## A.9 THE DETAILS OF VALIDITY OF VLM-BASED SIMILARITY SCORE

As this VLM-based similarity is newly introduced, its validity and robustness require evaluation. We validate this pipeline using a human-labeled partial-order set comprising 3,000 instances; each instance contains one real photo and four candidate 3D meshs $\{M_1, M_2, M_3, M_4\}$ with human similarity labels (e.g., $s_1 = s_2 > s_3 > s_4$). For each instance, our VLM pipeline produces a ranking of the four image–mesh pairs of each mesh based on VLM-based similarity introduced in Sec. 6.2.2 (e.g., $s_1 = 0.5, s_2 = 0.8, s_3 = 0.2, s_4 = 0.5$).We measure preference consistency as a hit if and only if the VLM ranking is identical to the human partial order; otherwise it is a miss. The protocol based on VLM-based similarity introduced in Sec. 6.2.2 achieves 92% agreement with human labels on this partial-order set, indicating high reliability for this similarity judgments.

## A.10 EXAMPLES OF FAILURE CASES

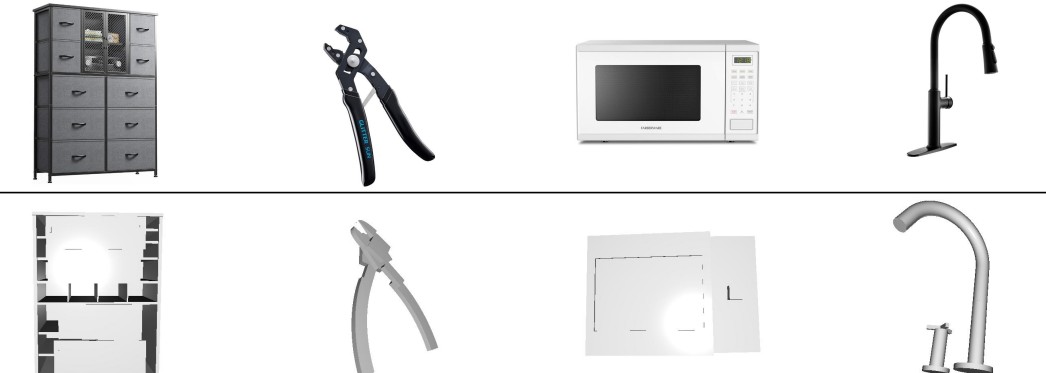

**Figure 8:** Examples of failure cases on real images.

A.11   EXAMPLES OF PERCEPTUAL EVALUATION BENCHMARK

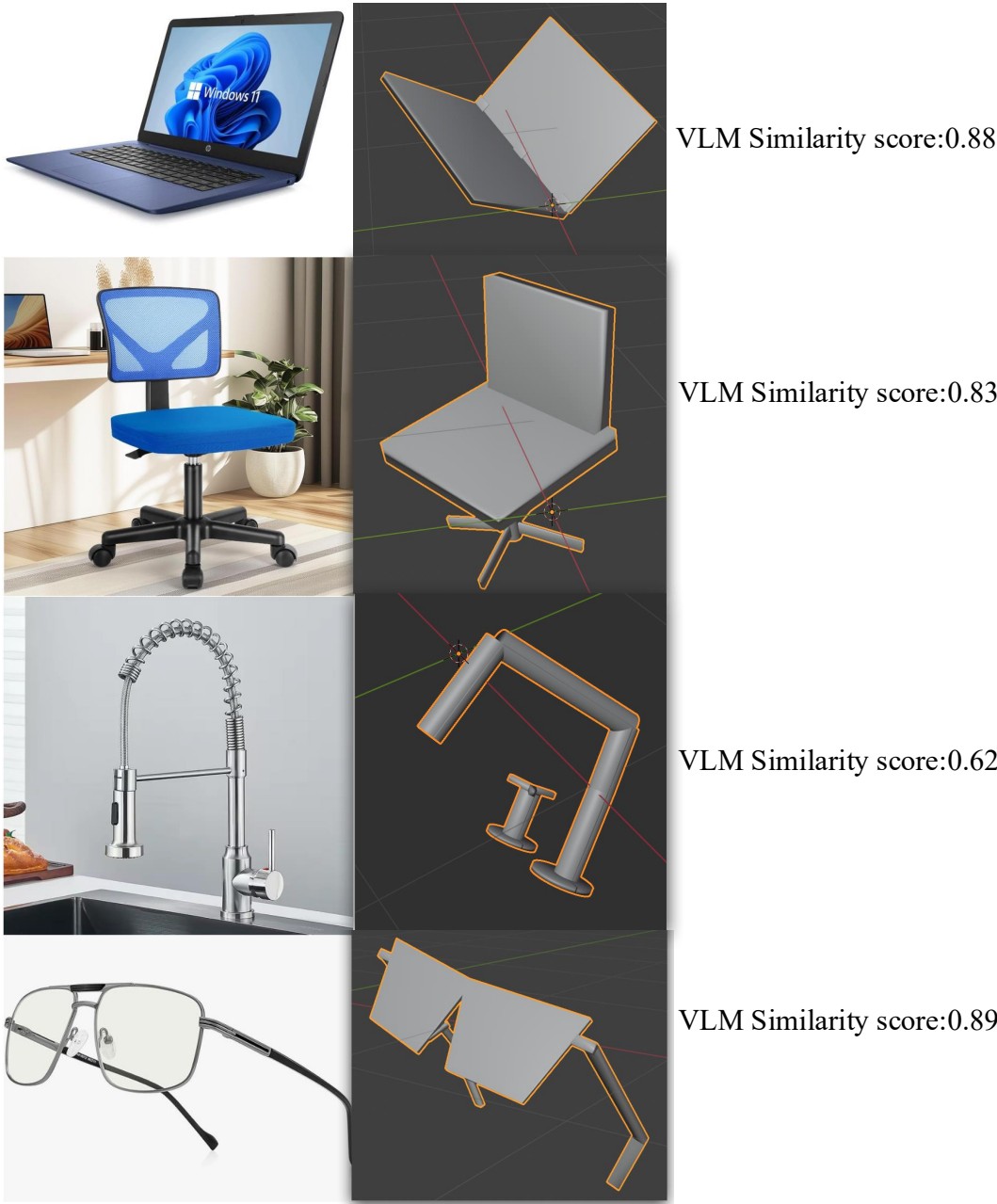

**Figure 9:** Examples of real images in our benchmark and its predicted mesh and VLM similarity score.

# B   RELATED WORK

## B.1   EXPLICIT OBJECT/SCENE REPRESENTATIONS.

Physics simulators such as MuJoCo use XML-based schemas (MJCF/URDF) to explicitly declare bodies, joints, inertias, contacts, and actuators; models are compiled from human-interpretable hierarchies into executable dynamics (Todorov et al., 2012). Programmatic CAD/DSLs (e.g., CSG, hierarchical part grammars) similarly encode geometry via primitives and parameters. These approaches offer editability and controllability but primarily target simulation/geometry specification

rather than analytic, concept-level structure. Digital Genes (Sun & Lu, 2025; Sun et al., 2024c;b) instead formalize analytic concepts—executable programs describing parts, parameters, and physical attributes intended to bridge perception, reasoning, and action .

### B.2 IMPLICIT NEURAL 3D REPRESENTATIONS.

Neural fields represent scenes or shapes as continuous functions learned from data, trading interpretability for fidelity. NeRF models radiance and density for photorealistic view synthesis (Mildenhall et al., 2020), while DeepSDF and Occupancy Networks learn signed-distance and occupancy functions for geometry modeling (Park et al., 2019). These methods excel at reconstruction and rendering but lack named parts, compositional parameters, or direct programmatic affordances, making them complementary to explicit Digital Gene code.

**Position of Digital Genes and GeneVLM.**   Compared to simulator XMLs (MJCF/URDF), Digital Genes are not merely scene descriptions but analytic programs emphasizing compositional parts, parameters, and functional attributes intended for both perception and control. Compared to implicit neural fields, they trade raw photorealism for interpretability and reusability. GeneVLM contributes an automatic image-to-gene pipeline that recovers such explicit programs from single images, advancing Digital Genes as a practical substrate for grounded reasoning and robotic manipulation (Sun & Lu, 2025; Sun et al., 2024c).

