# OpenReview forum: "GeneVLM: Automated Parsing Executable Digital Gene from a Single Image"
_ICLR.cc/2026/Conference — ICLR 2026 Conference Withdrawn Submission_

### Official Review · Reviewer_sjb1 · 2025-10-27

**Soundness:** 3
**Presentation:** 3
**Contribution:** 3
**Rating:** 6
**Confidence:** 4

**Summary:**

This paper proposes GeneVLM, a vision-language framework that automatically parses Digital Genes (structured, executable blueprints of 3D objects) from a single 2D image. Building on the Digital Gene concept introduced by Sun & Lu (2025), the work addresses a key bottleneck in structured 3D understanding: generating interpretable and executable object representations from everyday images.
Experiments show that GeneVLM reconstructs coherent object structures and scales predictably with model size, outperforming the Qwen-32B baseline.

**Strengths:**

1. The “image-to-Digital Gene” task is both original and practically important for interpretable robotic reasoning and structured 3D understanding.
2. Results demonstrate clear performance scaling from 7B to 32B parameters and reasonable generalization to real images.
3. The implementation details, evaluation metrics, and prompts are meticulously documented.

**Weaknesses:**

1. Despite the ColorImage-Gene dataset, the work relies heavily on synthetic data, and performance on complex real-world images remains limited.

2. The authors acknowledge that predicted parameters may lack the accuracy required for physical interaction, which limits immediate real-world applicability.

3. The newly proposed VLM-as-judge metric is interesting but somewhat subjective. Moreover, the geometric metrics could be expanded (e.g., IoU, part-level correspondence).

**Questions:**

1. How robust is GeneVLM to occlusions, cluttered backgrounds, and partial object visibility?
2. Could the constrained decoding FSM handle previously unseen templates or dynamic object categories?
3. Can the proposed method generalize to real-world data?

---

> ### Author Response · Authors · 2025-12-02
>
> # [1/4] Response to Reviewer sjb1
>
> > **Weakness1.** Despite the ColorImage-Gene dataset, the work relies heavily on synthetic data, and performance on complex real-world images remains limited.
>
> We acknowledge that synthetic data reliance is a limitation, but argue it is a **necessary trade-off** to enable million-scale training.
>
> ## Why Synthetic Data: Enabling Million-Scale Learning
>
> The reliance on synthetic data is not a weakness to be avoided but a deliberate design choice that makes our approach feasible. Without synthetic data generation, training a large model to parse Digital Genes from images would be impossible at the scale required for robust learning. The question is not whether to use synthetic data, but ***how to bridge the sim-to-real gap effectively***.
>
> ## Sim-to-Real Transfer
>
> We explicitly designed our training pipeline to address the synthetic-to-real gap through a staged approach.
>
> **Stage 1: WhiteImage Training (Geometric Prior Learning)**
>
> Data: 6M WhiteImage-Gene pairs (fixed-camera, no textures/lighting) + ~300K general SFT data
>
> Purpose: Learn the core task—mapping visual geometry to Digital Gene representations
>
> Why it works:
>
> - Visual encoder generalization: We use a **frozen pre-trained vision encoder**. This encoder has strong feature extraction capabilities that transfer to our synthetic WhiteImages.
> - Task-specific prompting: We use **different prompts to distinguish tasks** (e.g., “Parse the Digital Gene from this geometric rendering”), helping the model understand the domain.
>
> **Stage 2: ColorImage Fine-Tuning (Appearance Bridging)**
>
> Data: 800K ColorImage-Gene pairs (diffusion-generated with textures/lighting) + ~300K general SFT data
>
> Purpose: Bridge the appearance gap between synthetic and real images
>
> Why diffusion model helps:
>
> - Real-world visual information: The pre-trained diffusion model encodes rich real-world appearance priors (colors, textures, lighting) learned from large-scale real image datasets.
> - Appearance variation: We added a sketch constraint to the input of the diffusion model to ensure the structure remains unchanged during the generation process.
>
> **Mixing General SFT Data**
>
> Data: ~300K samples from InternVL-SFT, ShareGPT-4o, LLaVA
>
> Purpose: Preserve general visual understanding and real-image recognition capabilities
>
> This prevents the model from overfitting to synthetic distributions and maintains its ability to process real-world images. Our ablation studies (**Sec. 7.3.2**) demonstrate this dataset's effectiveness, with Model-WG (WhiteImage + general SFT) significantly outperforming WhiteImage-only on real images.
>
> Beyond ablations, our main experiments (**Sec 7.2**) demonstrate real-world applicability. Gene-32B significantly outperforms Qwen-32B few-shot baseline on real-world test images, showing learned capabilities transfer beyond synthetic training distribution. And performance improves consistently from Gene-7B to Gene-32B, suggesting the approach benefits from scale and is not fundamentally limited by synthetic data.
>
> ## Limitations and Future Directions
>
> We acknowledge remaining limitations and outline improvement directions.
>
> **Limitation**: Diffusion rendering can **inevitably alter mesh structure** (noted in **Sec. 8**), introducing noise into ColorImage-Gene pairs. This can negatively impact training if structural inconsistencies are significant.
>
> **Future improvements**:
>
> - Enhanced diffusion rendering: Improve structure-preserving constraints in the diffusion process to maintain better geometric consistency between WhiteImage and ColorImage.
> - Quality filtering: Implement automated quality checks to filter ColorImages with significant structural deviations.
>
> These are active research directions, and our current results demonstrate that the two-stage + SFT mixing strategy already achieves effective sim-to-real transfer.

---

> ### Author Response · Authors · 2025-12-02
>
> # [2/4] Response to Reviewer sjb1
>
> > **Weakness2.** The authors acknowledge that predicted parameters may lack the accuracy required for physical interaction, which limits immediate real-world applicability.
>
> ## Root Cause Analysis and Improvement Directions
>
> We identify two primary factors limiting current parameter accuracy:
>
> 1. **Information Insufficiency: Single 2D Image Constraint**
>
> The fundamental challenge is that single 2D images inherently lack the 3D information needed for precise geometric parameter estimation. Depth ambiguity, viewpoint dependency, and occlusions create an under-constrained problem where multiple 3D configurations can project to the same 2D observation. Natural extensions to address this include **video input** (temporal sequences provide multi-view information and motion cues for triangulation), **multi-view fusion** (even 2-3 views can dramatically reduce geometric ambiguity), or **hybrid 3D inputs** (incorporating depth maps or sparse point clouds while maintaining accessibility advantages). These directly address the root cause of geometric imprecision—insufficient information in single 2D observations.
>
> 2. **Vision encoder optimization mismatch**
>
> We use a **frozen pre-trained vision encoder**, which creates an important trade-off. Freezing is critical because the encoder’s strong feature extraction capabilities from billions of real-world images enable our model to generalize to real images despite training on synthetic data. Fine-tuning the encoder on our synthetic dataset would introduce strong biases and risk catastrophic forgetting, damaging the original capabilities that make our approach work. However, pre-trained encoders are optimized for **semantic understanding** (classification, captioning), not geometric precision—they may not encode fine-grained geometric details (exact angles, precise dimensions) as effectively as semantic information.
>
> To address this, we can explore **hybrid encoder architectures** that combine semantic-focused encoders with geometry-focused encoders like DINOv2 (which learns dense geometric features through self-supervision), using a learned fusion mechanism to leverage both.

---

> ### Author Response · Authors · 2025-12-02
>
> # [3/4] Response to Reviewer sjb1
>
> > **Weakness3.** The newly proposed VLM-as-judge metric is interesting but somewhat subjective. Moreover, the geometric metrics could be expanded (e.g., IoU, part-level correspondence).
>
> ## Validity of the VLM-as-a-judge Metric
>
> We acknowledge the concern that VLM-based metrics can appear subjective. However, we have rigorously validated this metric to ensure it serves as an objective proxy for human perception in real-world scenarios where 3D ground truth is unavailable.
>
> As detailed in **Section 6.2.2** and **Appendix A.9**, we conducted a dedicated validation experiment to prove the reliability of this metric. Specifically:
>
> - We constructed a dataset of **3,000 instances** (human-labeled partial-order sets).
> - We compared the rankings produced by our VLM pipeline against human rankings.
> - **Result:** The VLM-as-a-judge metric achieved a **92% agreement rate** with human annotators on this partial-order ranking task.
>
> This high level of consistency demonstrates that the metric is not merely subjective but is statistically aligned with human geometric perception, making it a reliable tool for evaluating semantic and geometric plausibility in the wild.
>
> ## Geometric Metrics and their Relationship to IoU/Part-Level Correspondence
>
> Regarding geometric evaluation, we wish to clarify that our benchmark already includes the standard metrics for single-image 3D reconstruction, and our Gene-level metrics explicitly cover part-level correspondence.
>
> - **Included Geometric Metrics (CD &** **F-Score****):** As shown in **Section 6.2.1** and **Table 1**, we report **Chamfer Distance (CD)** and **F-Score (@0.01, 0.02, 0.05)**. We selected these metrics following established protocols from recent state-of-the-art works (Liu et al., 2025; Huang et al., 2023; Liu et al., 2023b).
> - **Relationship with IoU:** While IoU is a common metric for voxel-based representations, CD and F-Score are generally preferred for mesh and point-cloud representations (which our Digital Genes produce via the rendering engine). CD and F-Score provide a continuous measure of surface alignment and are more sensitive to the fine-grained geometric details preserved by our parametric representation than voxel-IoU, which requires discretization and can suffer from resolution artifacts.
> - **Part-Level Correspondence:** We argue that "part-level correspondence" is centrally addressed by our novel **Gene-Level Evaluation (Section 6.1)**.
>     - Since a Digital Gene is inherently an explicit, structured representation of object parts (*ConceptInstances*), our **Concept Accuracy** metric (Eq. 3) functions as a strict **structural part-level correspondence metric**. It evaluates whether the model has retrieved the correct set of semantic parts (e.g., retrieving a "Curved_Handle" vs. a "Square_Handle").
>     - Furthermore, our **Float** **Error** metric evaluates the geometric precision of these specific parts.
>     - Therefore, our combined evaluation suite—Concept Accuracy (part existence/type), Float Error (part parameters), and CD/F-Score (global geometry)—provides a coverage that equals or exceeds the granularity of standard part-level correspondence or IoU metrics.

---

> ### Author Response · Authors · 2025-12-02
>
> # [4/4] Response to Reviewer sjb1
>
> > **Question1.** How robust is GeneVLM to occlusions, cluttered backgrounds, and partial object visibility?
>
> We evaluated GeneVLM’s robustness to occlusions, clutter, and partial visibility by constructing a controlled stress test on 500 real-world images. Using **Qwen-Image-Edit-Plus**, we synthetically added moderate occlusions, background clutter, and partial crops, then compared GeneVLM-7B/32B performance on these augmented images versus the *same clean originals*.
>
> **Results show that GeneVLM maintains strong performance under occlusion**, with only a modest drop in VLM geometric similarity:
>
> - **7B:** from **0.736 → 0.720** (−0.016)
> - **32B:** from **0.736 → 0.677** (−0.059)
>
> The degradation is expected because the training data contain **no occluded or truncated objects**, yet the models still preserve much of their structural reasoning ability. Qualitatively, GeneVLM continues to recover the major object parts and high-level geometry even when part of the object is obscured.
>
> Overall, these experiments demonstrate that GeneVLM exhibits **non-trivial robustness** to occlusion/clutter despite never being trained on such cases. Incorporating explicit occlusion augmentation in future data generation is expected to further improve performance.
>
> > **Question2.** Could the constrained decoding FSM handle previously unseen templates or dynamic object categories?
>
> Yes. Our FSM takes a **model-agnostic configuration file** as input, which defines the legal syntax rules for Digital Genes (valid template names, parameter dimensions, structural constraints). When extending to new templates or categories, we simply **update the configuration file**—no model retraining or FSM algorithm modification is required.
>
> For example, adding a new “Conical_Leg” template only requires: (1) defining it in the Digital Gene library, and (2) updating the FSM configuration to recognize the new template name and parameter structure. The FSM immediately enforces correct syntax for the new template.
>
> > **Question3.** Can the proposed method generalize to real-world data?
>
> Yes, the proposed method successfully generalizes to real-world data. We address the simulation-to-reality gap by fine-tuning on the photorealistic **ColorImage-Gene** dataset and mixing in general-purpose SFT data.
>
> **Evidence from the paper:**
>
> - **Real-World Benchmark:** We established a benchmark of **2,640 real-world images** (Section 6.2.2) to evaluate performance in uncontrolled environments.
> - **Quantitative Success:** In Table 1, using our proposed VLM-score (the validness of this VLM-score is discussed in section 6.2.2 last paragraph, and in previous response to review), the model shows ability to successfully generalizes to real-world data .
> - **Qualitative Robustness:** As demonstrated in **Figure 8 in Appendix**, GeneVLM correctly parses real photos (e.g., eyeglasses) across varying camera viewpoints, lighting conditions, and object articulations.

---

### Official Review · Reviewer_rpbJ · 2025-10-31

**Soundness:** 1
**Presentation:** 1
**Contribution:** 1
**Rating:** 0
**Confidence:** 5

**Summary:**

This paper introduces GeneVLM, a vision-language framework designed to parse a "Digital Gene" from a single 2D image. The "Digital Gene" is defined as an explicit, executable, and programmatic blueprint (represented as a JSON file) that encodes an object's hierarchical components and geometry.

To solve this, the paper devises three components:

1. A procedural data pipeline that programmatically synthesizes a large-scale (multi-million) dataset of image-gene pairs, starting from a "seed collection" of Digital Genes and applying augmentations.

2. The GeneVLM model, a VLM based on Qwen2.5-VL that incorporates a specialized float-to-token quantization scheme to handle numerical parameters .

3. A constrained decoding method that uses a Finite State Machine (FSM) to ensure the generated genes are syntactically valid and executable.

**Strengths:**

The paper is well-written, and the problem it aims to solve (2D image to a programmatic 3D representation) is clearly defined. The illustrations for the data pipeline (Figure 2) and model architecture (Figure 3) are clear and helpful for understanding the proposed system. The goal of inferring structured, interpretable, and executable 3D representations from single 2D images is highly significant for AI and robotics. The paper correctly identifies data acquisition as a major bottleneck for structured 3D models.

**Weaknesses:**

1. Vague and Unnecessary Terminology: The paper's core concept of a "Digital Gene" is a confusing neologism. The paper's own related work appendix (Appendix B) positions this concept against existing explicit representations like MJCF/URDF and programmatic CAD/DSLs . However, the paper's description of a "Digital Gene" as an "executable blueprint that encodes an object's hierarchical components, geometry, and functional attributes" sounds identical to these existing formalisms. The paper fails to substantially differentiate this concept, making the contribution feel like a re-branding of a known problem (image-to-procedural-model) with a strange and unnecessary term.

2. Fundamentally Handcrafted and Over-Engineered System: The work claims to present a learning framework, but the "learning" is almost entirely superseded by complex, handcrafted engineering. This system does not solve the "bottleneck" of gene creation; it just shifts it.

    i) Data Generation: The data pipeline is not based on real-world data. It starts from a "seed collection of Digital Genes" (which are presumably handcrafted) and then applies a complex "procedural pipeline" of "Rule-Based Augmentation" and "Stochastic Removal". This is an entirely synthetic, over-engineered process.

ii) Specialized Tokenization: The model requires a specialized, non-standard quantization scheme to convert floating-point numbers into single tokens. This is a significant engineering workaround that adds complexity and suggests the VLM architecture is poorly suited for regressing the continuous parameters that are essential to the "Digital Gene" format.

iii) FSM-Based Decoding: This is the most significant weakness. The paper admits that standard auto-regressive decoding "often fails to generate syntactically valid Digital Genes". The solution is to use a "constrained decoding method" that relies on a "Finite State Machine (FSM)" to "mask out all others [invalid tokens]" at every sampling step . This means the model is not learning the grammar or structure of the "Digital Gene"; it is being force-guided by a handcrafted parser. This undermines the entire premise of "parsing" and "generation."

3. Insufficient Baselines: The only baseline is a zero-shot Qwen-32B model, which is fed a complex prompt (Appendix A.4). This is a weak baseline for a highly specialized task. The paper does not compare its performance to any other established method for single-image 3D reconstruction or image-to-program synthesis. The custom "Digital Gene" format makes the benchmark insular and difficult to compare against.

**Questions:**

1. How does the "Digital Gene" representation fundamentally differ from existing programmatic representations like hierarchical part grammars or CAD/DSL formats, which are also explicit, hierarchical, and executable? The distinction seems minimal, making the central contribution feel like a re-branding of a known concept.

2. The reliance on an FSM for constrained decoding is a major weakness, as it implies the model never actually learns the syntax of a Digital Gene. Can the model produce syntactically valid genes without this handcrafted FSM? If not, how much of the "parsing" task is truly being learned versus being hard-coded into the inference logic?

3. The data pipeline (Fig. 2) requires a "seed collection of Digital Genes" and "Rule-Based Augmentation". How is this procedural synthesis approach more scalable or less of a "critical bottleneck" than the manual creation of 3D assets it claims to replace? It seems to have the exact same manual-effort bottleneck, just at the "seed" level.

4. The custom float tokenization scheme is a complex piece of engineering. What was the performance of a simpler, more standard approach, such as having the VLM output text-based floating-point numbers (e.g., "0.596") and letting the standard tokenizer handle them?

---

> ### Author Response · Authors · 2025-11-28
>
> # [1/3] Response to Reviewer rpbJ
>
> > **Question1.** How does the "Digital Gene" representation fundamentally differ from existing programmatic representations like hierarchical part grammars or CAD/DSL formats, which are also explicit, hierarchical, and executable? The distinction seems minimal, making the central contribution feel like a re-branding of a known concept.
>
> We thank the reviewer for this insightful question. We respectfully disagree that Digital Genes is merely a “re-branding” of existing representations. The fundamental distinction lies in the **level of abstraction**: Digital Genes operates at the **conceptual level** rather than the **geometric level**, enabling systematic knowledge definition, inferential reasoning, and seamless integration with downstream tasks—capabilities that CAD/DSL formats fundamentally lack.
>
> **Key Distinction 1: Representation Level—Geometric Construction vs. Concept Instantiation**
>
> - **CAD/DSL representations** focus on **geometric construction**: they describe *how to build a shape* through primitives (cuboids, cylinders), Boolean operations, and parametric constraints. Their primary goal is precise geometric specification for manufacturing or simulation.
>
> - **Digital Genes**, in contrast, operate at the **conceptual level**: they represent objects through ***ConceptTemplates*** and ***ConceptInstances*** that encode **analytic concepts**—decomposing objects into semantically meaningful, functionally grounded components. Beyond geometry, Digital Genes directly embed **semantics, functional properties, and manipulation knowledge** within the representation itself.
>
> This is not a syntactic difference but a **paradigm shift** from describing “what the geometry is” to encoding “what the object means and how it can be used.”
>
> **Key Distinction 2: Knowledge Definition and Systematic Propagation**
>
> This represents the most fundamental difference (demonstrated in ConceptFactory, NeurIPS 2024):
>
> - **In CAD/DSL**: Knowledge (e.g., “this is a handle,” “this region is graspable”) is **external**—it must be annotated and stored separately from the geometric representation, requiring manual labeling for each object instance.
> - **In Digital Genes**: Knowledge is **intrinsic**—defined once at the *ConceptTemplate* level and automatically propagated to object instances:
>
> ```
> Concept Template (define knowledge once)
>     ↓ (instantiation + parameterization)
> Concept Instance (inherits knowledge)
>     ↓ (point correspondence propagation)
> Specific Object (receives semantic/pose/affordance annotations)
> ```
>
> **Other Distinctions**
>
> - **Executability**: CAD/DSL programs are executable in the sense of **numerical computation**: executing the program generates meshes, computes volumes, or renders images. This is **generative executability**—producing geometric outputs. Digital Genes possess **inferential executability**: beyond generating geometry, they enable **reasoning and knowledge computation** at the conceptual level: pose computation (e.g., `grasp_pose = default_grasp_pose × handle_pose × object_pose`), and affordance inference (e.g., `is_graspable(x) = lid.is_handle(x) ∧ within_bounds(x)`).
> - **Differentiability**: Digital Genes are designed with **end-to-end differentiability**, this makes Digital Genes not merely a descriptive representation but a **learnable representation** that integrates seamlessly with modern deep learning pipelines—a critical capability for vision-to-manipulation systems that CAD/DSL formats lack.

---

> ### Author Response · Authors · 2025-11-28
>
> # [2/3] Response to Reviewer rpbJ
>
> > **Question2.** The reliance on an FSM for constrained decoding is a major weakness, as it implies the model never actually learns the syntax of a Digital Gene. Can the model produce syntactically valid genes without this handcrafted FSM? If not, how much of the "parsing" task is truly being learned versus being hard-coded into the inference logic?
>
> We conducted ablation experiments **without FSM-constrained decoding** on the same test set used in our geometric evaluation (Sec 6.2.1 in the paper). The metric here is **syntactic validity rate**—the percentage of generated outputs that can be successfully parsed as valid Digital Genes, regardless of semantic correctness.
>
> | Model               | Syntactic Validity (No FSM) |
> | ------------------- | :-------------------------: |
> | Qwen-32B (few-shot) |             86%             |
> | Gene-7B             |             95%             |
> | Gene-32B            |             97%             |
>
> **Key Observations**:
>
> - Our trained models achieve 95-97% syntactic validity without any FSM constraints, demonstrating clear evidence of learned syntactic knowledge.
> - The 11-point improvement of Gene-32B over the few-shot baseline (97% vs. 86%) shows that training on Digital Gene data enables the model to internalize syntax rules.
>
> These results directly refute the claim that “the model never actually learns the syntax.” The model demonstrably learns syntax; the FSM ensures production-grade reliability by handling the tail of the error distribution.
>
> Also, in code generation, enforcing syntax via constrained decoding **is a standard industry** and academic solution to ensure executability (e.g., JSON mode in GPT-4, grammar-constrained decoding in LLMs). At the same time, syntax is necessary but represents a small fraction of the learning challenge. The model’s primary achievement is learning to map visual observations to digital genes. Dismissing the model’s visual reasoning capabilities because we ensure valid JSON syntax is unreasonable.

---

> ### Author Response · Authors · 2025-11-28
>
> # [3/3] Response to Reviewer rpbJ
>
> > **Question3.** The data pipeline (Fig. 2) requires a "seed collection of Digital Genes" and "Rule-Based Augmentation". How is this procedural synthesis approach more scalable or less of a "critical bottleneck" than the manual creation of 3D assets it claims to replace? It seems to have the exact same manual-effort bottleneck, just at the "seed" level.
>
> We respectfully disagree that our approach has “the exact same manual-effort bottleneck.” The critical distinction lies in **where human effort is required** and **how it scales**. Traditional methods require per-object manual annotation that scales linearly with dataset size (O(N) for N objects). Our approach requires one-time seed design plus automated augmentation, where human effort is constant regardless of final dataset size (O(1) amortized cost). We demonstrate this with concrete numbers: our pipeline generates 6M training pairs from a seed collection, with human effort concentrated only in the initial seed design phase, not in the subsequent million-scale data generation.
>
> **Where Does Human Effort Actually Go?**
>
> Traditional 3D Asset Annotation (Per-Object Paradigm):
>
> - Human effort per object: Manual annotation of geometry, semantics, poses, affordances
> - Scaling behavior: Linear with dataset size—to create N annotated objects requires N × (annotation time)
> - Knowledge reuse: Minimal—each object must be annotated independently
>
> Our Procedural Synthesis (Seed + Augmentation Paradigm):
>
> - Human effort: One-time seed creation
> - Scaling behavior: Constant human effort—automated augmentation generates millions of variants
> - Knowledge reuse: Maximal—seed concepts are reused combinatorially
>
> The bottleneck is fundamentally different: traditional methods bottleneck on per-object human labor, while our method’s only constraint is one-time seed design plus computational resources (which are abundant and parallelizable).
>
> **Why Seed Collection is NOT the Same Bottleneck?**
>
> The reviewer’s concern assumes that “seed-level effort” scales similarly to “object-level effort.” This is incorrect for three reasons:
>
> - Seed quantity does not scale with dataset size. The seed collection size is determined by **category diversity**, not dataset size. Once you have seeds covering the relevant categories, you can generate arbitrarily large datasets through combinatorial augmentation—no additional human effort required.
> - Automated augmentation is truly automatic. Our rule-based augmentation (Sec. 4, Fig. 2) operates **entirely without human intervention**.
> - Knowledge reuse through *ConceptTemplates*. Our approach leverages the ConceptFactory framework (NeurIPS 2024), where knowledge is defined **once at the template level** and automatically propagates to instances.
>
> **Quantitative Evidence: Cost and Speed**
>
> We provide concrete numbers demonstrating our approach is less of a “critical bottleneck” and more scalable:
>
> **1. Per-Sample Annotation Cost**
>
> - **Traditional multi-knowledge annotation** (independent processes for each knowledge type): Part semantics annotation: ~8 min/object; Part pose annotation: ~10 min/object; Total: ~18 min/object for just two knowledge types (must repeat for each object)
> - **Our Digital Gene annotation** (unified, knowledge-reusable): Digital Gene annotation: ~7 min/object; All knowledge types (semantics, pose, affordances) captured in single annotation. Total: ~7 min/object.
>
> Our approach reduces manual cost by **61%** (7 min vs. 18 min) while capturing more knowledge types. As knowledge types increase, traditional methods require additional passes, while our cost remains constant.
>
> **2. Production-Scale Generation Speed**
>
> Our synthesis capability: **1M+ samples per day**, primarily CPU-based; minimal GPU usage (only for ColorImage diffusion stage)
>
> > **Question4.** The custom float tokenization scheme is a complex piece of engineering. What was the performance of a simpler, more standard approach, such as having the VLM output text-based floating-point numbers (e.g., "0.596") and letting the standard tokenizer handle them?
>
> We have **already performed this experiment exactly** in **Section 7.3.1 and Table 2**.
>
> - We compared `Model-S` (our method) against `Model-B` (standard tokenizer).
> - Our results explicitly show that the standard approach doubles the sequence length (Figure 7) and requires significantly more training compute for worse performance (Table 2).

---

### Official Review · Reviewer_wyfT · 2025-11-01

**Soundness:** 4
**Presentation:** 3
**Contribution:** 3
**Rating:** 6
**Confidence:** 4

**Summary:**

The paper introduces the image-to-Gene task and proposes GeneVLM, a vision-language model designed to parse executable digital genes—programmatic blueprints representing object structures—from a single 2D image. To train the model, the authors construct a dataset by modifying existing shape Genes using multiple strategies, resulting in both no-texture images and augmented color images via existing diffusion models. Experiments demonstrate the effectiveness of this dataset for the novel task. Extensive evaluations and ablation studies further validate the design choices in tokenization and training strategy.

**Strengths:**

* The image-to-Gene task is clearly motivated, aiming to produce a language-understandable representation of shapes from a single image.


* The dataset generation strategy is reasonable, and experiments demonstrate the usefulness of the generated data.


* The tokenization scheme and two-stage training strategy are well-designed, with ablation studies showing the contribution of each module.
* The proposed benchmark effectively demonstrates the method’s performance and capabilities.

**Weaknesses:**

* It is unclear whether all object categories are included in the training data, and whether the model can handle more complex instances or unseen categories. Including failure cases would help illustrate the limitations of the current approach.
* When generating synthetic training data, it is not clearly explained how modifications to shape parameters ensure the resulting shapes remain reasonable, or how the model determines which parts of a shape can be altered or deleted without invalidating the object.

**Questions:**

* In the dataset generation process, what information in the original Genes is used to support the different augmentation strategies?
* After applying these modifications, how do you ensure that the resulting shapes and structures remain reasonable and valid?

---

> ### Author Response · Authors · 2025-11-27
>
> We sincerely thank the reviewer for the insightful comments and constructive suggestions. Below, we provide detailed responses to each concern.
>
> > **Weakness1.** It is unclear whether all object categories are included in the training data
>
> We use a total of **32 object categories** in our training set, covering a wide range of common real-world objects. All categories used in evaluation are also included in the training data. The full list is: `Bottle, Box, Bucket, Chair, Clip, Dishwasher, Dispenser, Door, Eyeglasses, Faucet, Globe, Kettle, KitchenPot, Knife, Laptop, Microwave, Mug, Oven, Pen, Pliers, Refrigerator, Ruler, Safe, Shampoo, Stapler, StorageFurniture, Switch, Table, Trashcan, USB, Washingmachine, Window.`
>
>
> While it is infeasible to exhaustively cover all real-world object categories at the coarse **category** level, our dataset is designed to cover a richer set of fine-grained *ConceptTemplates* (e.g., handle, cover, button, etc.), which serve as the building blocks of real-world objects. This design allows the model to learn compositional object structures beyond mere category labels.
>
> > **Weakness1.** whether the model can handle more complex instances or unseen categories.
>
> To quantify object complexity, we propose two proxy metrics:
>
> - The number of parameters in *ConceptTemplates* (**Params**)
> - The average sequence length of *ConceptInstances* (**Avg Length**)
>
> We analyzed five representative categories, sorted by increasing complexity, and report their GENE-level evaluation results below:
>
> | category         | Params | Avg Length | Concept acc. | Float Err. |
> | :--------------- | :----: | :--------: | :----------: | :--------: |
> | Ruler            |   20   |    188     |    0.998     |   89.866   |
> | Bottle           |   28   |  260.504   |    0.982     |   61.530   |
> | KitchenPot       |  130   |  375.238   |    0.942     |  118.747   |
> | Chair            |  231   |  527.624   |    0.926     |  116.962   |
> | StorageFurniture |  744   |  3088.026  |    0.906     |  113.138   |
>
> The results show that although performance slightly decreases as object structural complexity increases, the overall performance remains stable, indicating good robustness to complex instances.
>
> For **unseen categories**, since our digital gene representation is a semantically rich textual language, the model is able to generate plausible digital genes for novel categories. However, these generated genes are not yet aligned with the **Rendering Engine** (Fig.1 in the paper), which limits their direct usability in downstream tasks. We plan to address this limitation by jointly learning to generate both the digital gene and its corresponding Rendering Engine in future work.
>
> > **Weakness1.** Including failure cases would help illustrate the limitations of the current approach.
>
> We have summarized the main limitations of our method in the ***Limitations and Future Work*** section. Following the reviewer’s suggestion, we will also include representative failure cases in the appendix of the revised version to provide a more transparent and comprehensive evaluation.
>
> > **Weakness2.** When generating synthetic training data, it is not clearly explained how modifications to shape parameters ensure the resulting shapes remain reasonable, or how the model determines which parts of a shape can be altered or deleted without invalidating the object.
>
> In the **Rule-Based Augmentation** stage, we modify gene parameters based on their intrinsic metadata, which are explicitly specified by the gene designer during the construction of the gene. This metadata includes:
>
> - Whether a parameter is discrete or continuous
> - Valid value ranges
> - Semantic constraints
> - ...
>
> To ensure the modified shapes remain valid and reasonable, we designed an exception handling mechanism that performs:
>
> - Self-intersection and mesh penetration checks between joints
> - Topology-based connectivity checks to ensure adjacent joints remain properly connected
>
> Only samples that pass all validity checks are retained.
>
> In the **Stochastic Removal** stage, we construct a *semantic dictionary*, which annotates each concept template as either: *Required* or *Optional*. Optional templates are randomly removed with a certain probability. Since this process only removes existing structures rather than creating new ones, it does not introduce mesh penetration or disconnection issues, as long as semantic constraints are satisfied.
>
> > **Question1.** In the dataset generation process, what information in the original Genes is used to support the different augmentation strategies?
>
> See **Weakness2.**
>
> > **Question2.** After applying these modifications, how do you ensure that the resulting shapes and structures remain reasonable and valid?
>
> See **Weakness2.**

---

### Official Review · Reviewer_eCVp · 2025-11-01

**Soundness:** 3
**Presentation:** 3
**Contribution:** 3
**Rating:** 6
**Confidence:** 4

**Summary:**

This paper studies a very important task of building structured world representations for robotic agents. Learning from those structured representations, agents can obtain better generalization ability. To have such representations, this paper studies recent digital gene, which gives concepts and instances some descriptions in JSON format. However, existing works have some scalability due to the need for a point cloud. Thus, this paper first studies how to construct such a representation from a 2D image. The authors proposed a whole pipeline of collecting data, finetuning a VLM, and evaluation. This is a very solid paper.

**Strengths:**

1. The problem studied in this paper is very important.
2. The content of the study is very solid. The authors provide a whole pipeline for generating a structured representation from 2D images, including dataset collection, fine-tuning VLMs, and evaluation.
3. The training data collected also considered the simulation to realism gap.

**Weaknesses:**

1. There are some issues with citation format in the paper. The authors use \citet for all citations. However, \citep should be used. For example, at line 173, it is "Our model, which we term GeneVLM, is built upon the Qwen2.5-VL Team (2025a) architecture."

**Questions:**

See weakness.

---

> ### Author Response · Authors · 2025-11-27
>
> Thank you for the constructive review. We appreciate your positive assessment of our problem setting and pipeline.
>
> Regarding the citation formatting issue: thank you for noting this. We will correct all improper uses of `\citet` and replace them with `\citep` where appropriate in the final version.

---

### Author Response · Authors · 2025-12-02
**Summary of reviews and rebuttal**

## Summary of reviews and rebuttal for Dear ACs, SACs, and PCs

Our paper introduces **GeneVLM**, a vision–language framework that parses Digital Genes—structured, executable, conceptual blueprints of objects—from a single 2D image, and **establishes the first multi-dimensional image-to-Digital-Gene benchmark**. Beside GeneVLM and benchmark, we **design an efficient and scalable procedural pipeline to synthesize a diverse, multi-million-pair dataset** of images and their corresponding Digital Genes.  Three reviewers (eCVp, wyfT, sjb1) give positive assessments (all 6s), emphasizing the importance of the task, the solidity of the pipeline, and the thorough experiments. One reviewer (rpbJ) gives a strong reject (0), mainly due to concerns about conceptual novelty and the amount of engineering. Below we summarize how we addressed these points.

**Minor presentation issue (Reviewer eCVp)**
 The only concern is incorrect use of \citet vs. \citep. We will fix all citation formatting in the camera-ready.

**Category coverage, complexity, and data validity (Reviewer  wyfT)**
 We clarified that our training set includes 32 object categories and that all evaluation categories are present in training. We introduced two complexity measures (template parameter count and instance sequence length) and showed that performance degrades only slightly as structural complexity increases, indicating robustness to complex instances. We explained how gene metadata, validity checks, and required/optional part semantics ensure augmented shapes remain reasonable, and we will add failure cases in the appendix.

**Novelty, engineering, and scalability (Reviewer  rpbJ)**
 We explained that Digital Genes differ from CAD/DSL and part grammars by operating at a conceptual level: they encode semantics, functions, and manipulation knowledge via ConceptTemplates/Instances and support knowledge propagation and inferential executability, not just geometry generation.
 We showed that syntax is learned rather than hand-coded: without FSM constraints, syntactic validity is 86% for Qwen-32B few-shot, vs. 95% (Gene-7B) and 97% (Gene-32B), demonstrating that the model internalizes Digital-Gene grammar, while the FSM is used only to guarantee production reliability—analogous to grammar-constrained code generation or JSON modes in modern LLMs.
 On scalability, we argued that traditional methods require per-object annotation (O(N) effort), whereas our approach uses one-time seed design plus fully automatic augmentation to generate 6M pairs, making human effort essentially O(1) w.r.t. dataset size. We quantified that Digital Gene annotation is substantially cheaper per object while capturing more types of knowledge.
 Finally, our specialized float tokenization is empirically justified: in the paper we compare against standard text floats and show that the naive approach doubles sequence length and yields worse performance for higher compute.

**Synthetic data, real-world generalization, and metrics (Reviewer sjb1)**
 We motivated synthetic data as necessary for million-scale training and described our sim-to-real strategy: (1) large-scale WhiteImage-Gene training for core geometry, (2) ColorImage-Gene fine-tuning via diffusion-rendered images, plus (3) mixing ~300K general SFT samples to preserve real-image competence. We built a 2,640-image real-world benchmark and showed GeneVLM significantly outperforms Qwen-32B and benefits from scaling (7B → 32B).

 We acknowledged that single 2D images inherently limit parameter accuracy for physical interaction, and outlined natural extensions (multi-view, video, depth, hybrid encoders combining semantic and geometric features).
 For evaluation, we validated our VLM-as-judge metric on a 3,000-instance human-labeled preference dataset, achieving 92% agreement with human rankings, and we already report standard geometric metrics (Chamfer Distance, F-Score) suitable for mesh/point-cloud outputs. Our Gene-level metrics (Concept Accuracy + Float Error) directly capture part-level correspondence and part geometry.

**Robustness and extensibility (Reviewer  sjb1)**
 We ran a controlled stress test on 500 real images with synthetically added occlusion and clutter; GeneVLM’s similarity scores drop only modestly, showing non-trivial robustness despite training on non-occluded data. The FSM is driven by a configuration file, so adding new templates or categories only requires updating this config, not retraining the model.

---
In summary, three reviewers view the work as solid and above the acceptance threshold, and their concerns are addressed with concrete clarifications and additional analyses. The remaining 0-score review largely reflects disagreement over framing and the role of engineering rather than issues of correctness. We believe the rebuttal demonstrates that the work is sound, novel in its representation and benchmark.

---

### Note · Authors · 2026-01-29

I have read and agree with the venue's withdrawal policy on behalf of myself and my co-authors.

---

### Meta-Review · Area_Chair_1zrE · 2026-01-07

**Summary:**

The paper received dramatic initial reviews of 6-6-6-0, making it a rather difficult case. Upon looking into the reviews, here's my findings:
1. All 3 positive reviewers who gave borderline accept are not very detailed. They slightly lean positive but also pointed out limitations of the paper and seem rather unsure.
2. The 1 negative reviewer provided the most detailed review among all, though with an extremely negative opinion. The authors' note has been considered thoroughly. I agree that the review is indeed not perfect, e.g. overlooking an experiment already provided in the text as well as giving unfair judgement on the presentation quality. However, the overall merit of the judgements remains valid and important. Please see "Reviewer Concerns" for more details, but after careful consideration, I do believe that the raised concerns are valid and have not been sufficiently addressed by the rebuttal, so I recommend a Reject.

**Reviewer Concerns:**

I will discuss the details of the strongly negative reviews and the rebuttal here.

Concerns reviewer raised:
1. "Digital Gene" is a rebranding of existing concepts like URDF, MJCF, or programmatic CAD. This raises suspicion for overclaiming and brings confusion for future readers.
2. Heavy reliance on handcrafted engineering including several aspects of the pipeline: data, FSM decoding, and tokenization. This raises strong concerns for scalability and generalization ability in real-world performance.
3. Baselines used in the paper is weak and unfair because zero-shot Qwen model has not been established as a single-image 3D reconstruction or image-to-program synthesis model. Without fair baselines it's hard to evaluate the significance of contribution.

The authors' rebuttal:
1. They argue "Digital Genes" operate at a conceptual level, not a geometric one. Unlike CAD (which describes how to build shapes), Digital Genes encode semantics, functional affordances, and "inferential executability" (reasoning about grasping/usage) inherently. They also claim Digital Genes are end-to-end differentiable, unlike standard CAD formats.
2. They argue the "seed" approach fundamentally changes the scaling laws of annotation. Traditional annotation is linear ($O(N)$—more data needs more humans), whereas their approach is amortized constant time ($O(1)$—design one seed, generate millions of variants). They cite a 61% reduction in annotation time per object.
3. The authors pointed to internal ablations studies.

My opinion:
1. This author's proposed distinctions imply that URDFs cannot hold metadata, which is false. Adding semantic tags or custom property fields to a standard URDF or XML structure is trivial. The authors have not proven that a new file format was necessary to achieve "conceptual" reasoning. The reviewer is likely to view the "Digital Gene" as simply a JSON schema wrapper around standard scene graph concepts. I tend to agree with the reviewer that the term is leaning more towards a "re-branding".
2. IMO this is the biggest weakness of the proposed method. The author's argument of O(1) vs. O(N) complexity will stand if there's no large-scale 3D dataset in the world, which is clearly not the reality. Additionally, they did not address the reviewer's implication that the system is trained entirely on "rule-based" synthetic images. A model trained on 6 million procedurally generated data often suffers catastrophic domain shift when applied to real data. The rebuttal focuses on annotation cost rather than visual realism or sim-to-real transfer, which is the actual bottleneck for VLM robotics deployment.
3. The concerns regarding the lack of comparison against SOTA methods such as "NeRF-based reconstruction, primative-based fitting, and other VLM-to-3D pipeline, still remain. Ablations studies along is not enough for justifying the significance of contribution.

**Reviewer Scores:**

Reviewer rpbJ's initial review is rather harsh. I believe a 2 instead of 0 is more fair. And the presentation score should be higher than 1. Yet, the rebuttal provided by the authors seems to be diverting from the concerns instead addresssing them head-on. I think reviewer rpbJ raised valid and important points in the initial review and it doesn't look like he would've change his/her opinions after seeing the rebuttal.

In the meantime, other reviewers gave borderline reviews and raised similar concerns as reviewer rpbJ.

---

### Decision · Program_Chairs · 2026-01-26

Reject